

# Forage yields and nutritive values of oat and triticale pastures for grazing sheep in early spring

Hülya Hanoğlu Oral

Department of Animal Production and Technologies, Faculty of Applied Sciences, Muş Alparslan University, Muş, Türkiye

## ABSTRACT

**Background**. Small-grain winter cereals can be utilized as early spring pastures in temperate climates to relieve grazing pressure and potentially mitigate feed shortages. This study was conducted to determine the effects of triticale and oat cereal pastures grazed by sheep during early spring on forage yields, nutritive values, and nutritional requirements of sheep.

**Methods**. The research was carried out over three consecutive years, from 2015 to 2017, at the Sheep Research Institute in Bandırma-Balıkesir, located in the Marmara region of Türkiye. The treatments were arranged in a completely randomized block design, with the two forage species, triticale and oat, randomized within each of three blocks. The animal material for the study consisted of 24 Karacabey Merino sheep, each 2 years old, with an average live weight of $57.6 \pm 0.5$ kg, all in the late lactation stage. In each replication, four sheep were included, resulting in a total of 12 sheep grazing in each of the triticale and oat pastures. The sheep grazed exclusively on the cereal pastures without any additional feed, and had unrestricted access to water throughout the entire period of the experiment. The dry matter yields (DMY), dry matter intakes (DMI), nutritive values, and mineral contents of the cereal species were determined.

**Results**. The DMY showed significant differences over the years ($P < 0.05$). No differences in DMY were observed between pastures, with oats yielding 11.99 t ha$^{-1}$ and triticale yielding 11.08 t ha$^{-1}$. During the grazing period, the change in DMY was significant in all years ($P < 0.05$). The average DMI of the sheep was 2,003.5 g d$^{-1}$ for triticale and 2,156.6 g d$^{-1}$ for oat, respectively, and DMI exhibited no significant differences across pastures. Although there was no difference in DMI between 2015 and 2016, the lowest consumption occurred in 2017 ($P < 0.05$). Additionally, while DMI showed different trends each year based on the periods, it generally decreased by the end of the grazing period. While both pastures provided similar nutritive values, significant differences were observed in the crude protein (CP), acid detergent fiber (ADF), *in vitro* true DM digestibility (DDM), and metabolisable energy (ME) values across the years. Over the years, as the grazing period progressed, CP levels decreased while neutral detergent fiber (NDF), ADF, and acid detergent lignin (ADL) increased, resulting in reduced DDM and ME values. The phosphorus (P) content in triticale was higher than in oats, but there were no differences in the content of other minerals between them. Between the years, significant differences were observed in the levels of phosphorus (P) and iron (Fe), while changes in other elements were insignificant. The variation in mineral content during the grazing process differed over the three years. Study results indicated that the nutritional values of triticale and oat pastures

Corresponding author
Hülya Hanoğlu Oral,
h.hanoglu@alparslan.edu.tr

are similar, and both can effectively be used to provide sufficient feed to meet the early spring forage requirements for sheep.

## INTRODUCTION

In livestock production systems based on grazing, the impact of feed shortages on animal production is particularly severe during winter and early spring. Therefore, identifying high-quality forage resources for grazing livestock in early spring is crucial both for sustainable animal production and to prevent the degradation of perennial pastures due to early grazing (*Gökkuş & Hanoğlu Oral, 2022*). Exploring different forage options can extend the seasonal grazing period beyond the capacities of perennial pastures (*Phillips et al., 2021*), thereby alleviating grazing pressure and potentially mitigating the feed shortage issue. In this context, cereal pastures could serve as an effective solution to address feed shortages, offering a viable alternative to mitigate this challenge (*Harrison et al., 2011*). This approach not only addresses the gap in feed availability (*Lauriault et al., 2022*) but also promotes growth in sheep (*Masters & Thompson, 2016*) and alleviates pressure on rangelands by preventing early spring grazing (*Chen et al., 2016*), enabling farmers to postpone grazing on perennial pastures (*Thomas et al., 2014*; *Phillips et al., 2021*).

Small grain winter cereals initiate early growth in spring, allowing them to reach grazing maturity sooner (*Torell et al., 1999*). Their high forage quality (*Lemus, 2017*) and rapid regrowth after grazing make them ideal for establishing rotational pastures (*Gökkuş & Hanoğlu Oral, 2022*). Given that cereal pastures provide high-quality forage, they effectively minimize the additional energy and protein requirements for animals (*Luis et al., 2020*). The nutritional content of cereal crops is particularly high during their initial growth stages, meeting early spring requirements for superior quality feed. This offers benefits similar to those of concentrated feed (*Coblentz & Walgenbach, 2010*), providing an excellent alternative for small ruminant producers. Utilizing cereal forages for grazing from the early vegetative stage through the advancing growth stage holds considerable promise. It can extend the grazing season and reduce supplemental feed expenses for these producers (*Rihawi et al., 2010*). These forages are often compatible with the high nutritional requirements of ewes during pregnancy and lactation (*Masters & Thompson, 2016*). However, concerns exist regarding potential mineral imbalances during the vegetative growth stage of cereal crops (*Masters et al., 2019*). Mineral nutrition disorders, such as hypocalcemia, hypomagnesemia, or osteodystrophies, can greatly decrease the productivity and profitability of grazing systems (*Dove, Masters & Thompson, 2016*).

All small grain winter cereals are suitable for grazing, but oats are often prioritized for their rapid growth and good post-grazing recovery, which is especially useful in establishing short-term artificial pastures (*Coblentz et al., 2013*; *Coblentz & Gildersleeve, 2014*). Forage

oats offer numerous advantageous ecological and physiological traits, including high yields, elevated levels of crude protein, abundant water-soluble carbohydrates, and significant amounts of digestible neutral dietary fiber, making them a superior feed option due to their high palatability (*Coblentz et al., 2012*). Furthermore, attributes such as their relatively brief growth cycle, strong resilience to abiotic stressors, and versatility have contributed to their widespread recognition as valuable forage crops (*Zhang et al., 2023*). Similarly, triticale significantly contributes to animal production by providing sufficient and high-quality forage (*Glamočlija et al., 2018*; *Bumbieris Junior et al., 2020*; *De Zutter et al., 2023*). In fact, due to its high forage capacity and preference under grazing conditions, triticale enhances animal production more than common vetch (*Tölü et al., 2013*), making it more advantageous compared to barley and rye pastures (*Keles et al., 2016*). Research on the feeding value of small grain winter cereals also reveals their effects on the performance of cattle (*Myer et al., 2011*; *Mullenix et al., 2014*; *Dubeux Jr et al., 2016*; *Phillips et al., 2021*; *Lauriault et al., 2022*), lambs (*Dove et al., 2002*; *Ates et al., 2015*; *Keles et al., 2016*), and goats (*Tölü et al., 2013*; *Akbağ, 2022*) as potential forage crops. However, the literature regarding their effects on sheep performance is limited, with only a few studies available (*Dove & McMullen, 2009*; *Vidyarthi et al., 2011*; *Dove & Kelman, 2015*).

For this study, triticale (*X Triticosecale Wittmack* L.) and oats (*Avena sativa* L.), known for their superior feeding value in many regions of Türkiye and their potential as forage crops for pastures, due to their adaptation to low temperatures, were selected. Our hypothesis was that for forage crops grazed in early spring, both the forage yield and nutritive value would be influenced by the species and the timing within the grazing interval, leading to variations in meeting the nutritional requirements of grazing sheep. The objective of this study was to determine the forage yield and nutritional quality of triticale and oats as cereal pastures for sheep grazing throughout the entire early spring grazing period, and to evaluate the extent to which these pastures meet the nutritional requirements of the sheep.

## MATERIALS & METHODS

The Animal Experiments Ethics Committee of Çanakkale Onsekiz Mart University approved all aspects of animal care management for this study, including handling, housing, and feeding procedures (protocol number: B.30.2. ÇAÜ.0.05.06-050.04/82, approval date: August 27, 2014).

### Site and establishment of the cereal pastures

The research was conducted at the Sheep Research Institute in Bandırma-Balıkesir, ($40°17'16''$ to $40°20'17''$ northern latitude and $27°53'37''$ to $27°58'25''$ eastern longitude) located in the Marmara region of Türkiye, over three consecutive years from 2015 to 2017. Following cultivation and seedbed preparation, triticale (*X Triticosecale Wittmack*, cv. Karma) and oats (*Avena sativa* L., cv. Faikbey) were sown in 1,080 $m^2$ plots using a commercial seed drill at the end of October. The plot area was determined based on the maintenance dry matter requirement of the sheep (*Holechek, Pieper & Herbel, 2004*). The plots, enclosed by a fence, were divided into two sections $-600$ $m^2$ and 480 $m^2$—for

rotational grazing. The treatments were organized using a completely randomized block design, with the two forage species randomly assigned within each of the three blocks. Typically, it is recommended to sow 200 kg ha$^{-1}$ of seeds for triticale (*Geren et al., 2012*) and 150 kg ha$^{-1}$ for oats (*Mut, Demirtaş & Köse, 2021*). However, in the experiment, to create a dense stand, these amounts were approximately increased by 50%, resulting in 300 kg ha$^{-1}$ for triticale and 200 kg ha$^{-1}$ for oats.

The site soils were generally neutral and non-saline, with low to moderate lime content, and predominantly clayey or clayey-loamy in texture. The soil contained moderate organic matter. Phosphorus (P) levels ranged from low to sufficient. Potassium (K) and magnesium (Mg) levels were sufficient to excess. Calcium (Ca) was in excess. Copper (Cu) and manganese (Mn) levels were sufficient. Iron (Fe) levels were in excess. Zinc (Zn) levels ranged from low to sufficient. Based on the soil analysis results, 200 kg ha$^{-1}$ of diammonium phosphate was applied at the time of sowing, and 200 kg ha$^{-1}$ of ammonium sulfate fertilizer was applied before the onset of grazing at the end of February. As a result, from the time of sowing to the beginning of grazing, a total of 80 kg ha$^{-1}$ of nitrogen (N) and 90 kg ha$^{-1}$ of phosphorus pentoxide ($P_2O_5$) were applied (*Akcura & Ceri, 2011*).

## Climate Characteristics

The monthly rainfall and average mean temperature during the experiment are presented in Table 1. The long-term average annual total precipitation was 643.8 mm, with an average monthly temperature of 14.2 °C. However, the annual total precipitation for 2015, 2016, and 2017 was 546.3 mm, 766.9 mm, and 662.4 mm, respectively. Thus, the precipitation in 2015 was below the long-term average, while it exceeded this average in 2016 and 2017. Despite the high precipitation in 2016, a significant portion of it (601.4 mm, 78.4%) occurred during January, February, March, November, and December, leaving the spring and autumn months, which are critical for plant growth, typically dry. The average temperatures during the trial years were generally above the long-term average, with recorded temperatures of 15.3 °C in 2015, 15.7 °C in 2016, and 14.7 °C in 2017, the last of which was close to the long-term average.

## Forage yield of cereal species

To determine both the dry matter yields (DMY) and the dry matter intake (DMI), protective enclosures measuring 6.25 m$^2$ (2.5 × 2.5 m) were established in each plot using fencing material. The DMY (t ha$^{-1}$) was determined by selecting three random 1 m$^2$ quadrats from each of the six plots. Each quadrat was cut using sickles to a stubble height of 50 mm at approximately 20-day intervals throughout the experiment period. All cuts from the quadrats were weighed, sampled, and dried to calculate the DMY. The DMI (g day$^{-1}$) of the sheep was determined by subtracting the yield within the protected area from that of the grazed area. To illustrate plant development across two sampling periods, the positions of the cages were alternated each time (*Gökkuş, Koç & Çomaklı, 2009*).

## Animal experiment and procedures

The animal experiment involved 24 Karacabey Merino sheep (95% German Mutton Merino × 5% Kıvırcık), all 2 years old, with an average live weight of 57.6 ± 0.5 kg, in late
**Table 1** Climate data recorded at Sheep Research Institute, Bandırma-Balıkesir, Türkiye, during the experiment period.[1]

| | Total rainfall (mm) | | | | Average temperature (°C) | | | |
|---|---|---|---|---|---|---|---|---|
| | LY[*] | 2015 | 2016 | 2017 | LY[*] | 2015 | 2016 | 2017 |
| January | 78.1 | 88.2 | 133.2 | 167.0 | 5.3 | 5.9 | 5.5 | 2.5 |
| February | 80.3 | 98.8 | 104.4 | 26.8 | 5.9 | 6.4 | 10.5 | 7.1 |
| March | 58.0 | 42.6 | 101.0 | 55.6 | 7.7 | 8.7 | 10.7 | 9.4 |
| April | 46.1 | 80.2 | 19.3 | 24.0 | 12.1 | 11.7 | 15.7 | 11.9 |
| May | 24.1 | 11.8 | 36.4 | 28.8 | 16.6 | 18.6 | 18.0 | 17.2 |
| June | 24.3 | 26.2 | 24.8 | 25.6 | 21.2 | 21.4 | 23.4 | 22.7 |
| July | 6.3 | 0.0 | 15.7 | 25.4 | 23.7 | 24.8 | 25.3 | 24.4 |
| August | 9.2 | 5.0 | 0.6 | 43.8 | 23.8 | 26.2 | 25.5 | 25.0 |
| September | 71.2 | 19.6 | 57.9 | 42.8 | 20.4 | 23.3 | 21.7 | 21.5 |
| October | 94.5 | 105.7 | 10.8 | 57.4 | 15.8 | 17.1 | 16.3 | 14.3 |
| November | 66.6 | 50.6 | 148.6 | 60.4 | 11.1 | 13.3 | 11.3 | 10.6 |
| December | 85.1 | 17.6 | 114.2 | 104.8 | 7.3 | 6.4 | 4.0 | 9.6 |
| Total/Average | 643.8 | 546.3 | 766.9 | 662.4 | 14.2 | 15.3 | 15.7 | 14.7 |

**Notes.**
[1] The data was obtained from the Balıkesir-Bandırma Meteorology Directorate.
[*] LY: Long years (1950–2017).

lactation. Each year, twelve sheep were allocated to a treatment group and each group was divided into three subgroups containing four sheep, resulting in a total of 12 sheep grazing in each of the triticale and oat pastures. The sheep were randomly distributed among the experimental groups based on their weights at the beginning of the experiment. Grazing commenced when plants reached a height of 20–25 cm (*Shewmaker & Bohle, 2010*; *Smith et al., 2011*). Grazing periods were as follows: 37 days from April 1st to May 7th in the first year, 42 days from April 1st to May 12th in the second year, and 38 days from April 11th to May 18th in the third year. The sheep grazed exclusively on the cereal pastures and were not provided with additional feed, and had unrestricted access to water throughout the entire period of the experiment.

## Analytical procedures

Forage samples collected were dried in an air-forced oven at 60 °C for at least 48 h until a constant weight was achieved. These dried samples were then ground to pass through a 1-mm screen using a Retsch mill (Retsch GmbH, Haan, Germany) and analyzed for dry matter (DM, 935.29), ash (942.05), and crude protein (CP, 954.01) according to *AOAC (2003)* methods. Determinations of neutral detergent fiber (NDF) and acid detergent fiber (ADF) followed the methods described by *Van Soest, Robertson & Lewis (1991)*, using an Ankom200 Fiber Analyzer (Ankom Technology, Fairport, NY, USA). The acid detergent lignin (ADL) content in ADF samples was determined by immersing them in 72% sulfuric acid for 3 h in beakers. The NDF analysis included the use of heat-stable amylase and sodium sulfite, and both NDF and ADF measurements included residual ash.

*In vitro* true DM digestibility (DMD) was assessed using the Ankom Daisy II incubator (Daisy II; Ankom Technology, Fairport, NY, USA) by incubating samples for 48 h. Ruminal fluid was sourced from a non-pregnant sheep fed a 60:40 forage-to-concentrate ratio every

three days. The final bag weight after NDF analysis was recorded and used to estimate digestibility. The DMD value was calculated as follows:

DMD (%) = 100 −[(final weight (filter bag + sample) −(weight of the filter bag × correction factor for a blank filter bag)) ×100] / (weight of the sample × percentage of dry matter in the feed).

The DMD values were utilized to estimate digestible energy (DE, kcal kg$^{-1}$) employing the regression equation reported by *Fonnesbeck et al. (1984)*: DE (Mcal kg$^{-1}$) = 0.27 + 0.0428 (DMD%). Subsequently, DE values were transformed into metabolisable energy (ME) using the formula reported by *Khalil, Sawaya & Hyder (1986)*: ME (Mcal kg$^{-1}$) = 0.821× DE (Mcal kg$^{-1}$).

Mineral analyses for P, K, Ca, Mg, Fe, Zn, and Mn were conducted using an Inductively Coupled Plasma Spectrometer (Perkin-Elmer, Optima 2100 DV, ICP/OES, Shelton, CT, USA), following the method described by *Mertens (2005)*.

### Statistical analysis

The agronomical data and the nutritive value of the forages were analyzed by ANOVA, using a model that included year, pasture type (triticale, oat), and the interaction between year and pasture type, within a completely randomized block design. Due to variations in the number of grazing days each year, the data for each period were analyzed by year. The Duncan multiple comparison test was applied to compare the means. Statistical analysis of the data was conducted using the *SAS (1999)* statistical package.

## RESULTS

### Dry matter yield and dry matter intake

The average DMY and the DMI from cereal pastures were presented in Table 2. Notably, the DMY of cereal pasture species showed significant annual variations ($P < 0.05$). Rainfall influenced the DMY, reaching its highest level in 2016. The average DMY of the oat and triticale were 11.99 t ha$^{-1}$ and 11.08 t ha$^{-1}$ respectively, and no significant difference was found between the cereal pasture species in terms of the DMY. The interaction between treatments on the DMY was significant ($P < 0.05$). The effect of the year on the DMI was significant ($P < 0.05$), whereas the effect of the cereal pasture species was insignificant ($P > 0.05$).

The DMY and the DMI from cereal pastures by periods were presented in Table 3. Due to differences in the number of sampling time each year, data were analyzed by year. During the grazing seasons of the years, DMY varied significantly depending on the sampling time. In 2015, DMY showed a different variation compared to other years. In this year, the DMY decreased between April 1st and April 15th and between April 29th and May 7th. The highest DMY was achieved on April 29th, 2015, and April 28th, 2017, while the lowest values were observed at the end of the grazing season ($P < 0.05$). During the first, second, and third years of the study, oats and triticale produced similar amounts of forage. The DMI decreased towards the end of the grazing periods in 2015 and 2017 ($P < 0.05$). In the second year, the highest DMI was recorded because the DMY was greater than in other years, and the yield of the oat pasture was higher than that of the triticale pasture (Fig. 1).

**Table 2  Dry matter yield and dry matter intake from cereal pastures.**

| Variable | Year | | | Pasture | | P values | | |
|---|---|---|---|---|---|---|---|---|
| | 2015 | 2016 | 2017 | Oat | Triticale | Year | Pasture | Year*Pasture |
| DMY (t ha$^{-1}$) | 10.15$^b$ ± 0.50 | 14.78$^a$ ± 0.50 | 9.67$^b$ ± 0.50 | 11.99 ± 0.41 | 11.08 ± 0.41 | 0.0001 | 0.1459 | 0.0240 |
| DMI (g day$^{-1}$) | 2,331.4$^a$ ± 207.07 | 2,653.3$^a$ ± 180.60 | 1,255.6$^b$ ± 180.60 | 2,156.6 ± 147.50 | 2,003.5 ± 147.50 | 0.0007 | 0.4800 | 0.4700 |

Notes.
DMY, dry matter yield; DMI, dry matter intake.
The results in the table are presented as least square means ± standard errors.
a,b,c, the difference between the mean values expressed with different letters in the same treatment group is statistically significant.

**Table 3  Dry matter yield and dry matter intake from cereal pastures by periods.[1]**

| Sampling time | | | Dry matter yield (t ha$^{-1}$) | | | Dry matter intake (g day$^{-1}$) | | |
|---|---|---|---|---|---|---|---|---|
| 2015 | 2016 | 2017 | 2015 | 2016 | 2017 | 2015 | 2016 | 2017 |
| | | | Period | | | | | |
| 1 April | 1 April | 11 April | 3.27 b | 6.20 a | 2.77 b | – | – | – |
| 15 April | 21 April | 28 April | 0.55 c | 4.78 ab | 4.63 a | 2,457.3 a | 2,231.8 b | 1,847.2 a |
| 29 April | 12 May | 18 May | 5.74 a | 3.79 b | 2.27 b | 2,587.7 a | 3,036.7 a | 1,126.7 b |
| 7 May | – | – | 0.58 c | – | – | 1,568.6 b | – | – |
| SEM | | | 0.162 | 0.504 | 0.439 | 232.80 | 210.10 | 180.30 |
| | | | Pasture | | | | | |
| Oat | | | 10.05 | 16.59 | 9.33 | 2,419.2 | 2,825.0 | 1,530.7 |
| Triticale | | | 10.24 | 12.97 | 10.02 | 2,161.8 | 2,481.8 | 1,367.3 |
| SEM | | | 0.456 | 1.326 | 1.074 | 190.10 | 210.10 | 180.3 |
| | | | P-value | | | | | |
| Pasture | | | 0.7738 | 0.0649 | 0.6575 | 0.6260 | 0.3038 | 0.5656 |
| Period | | | 0.0001 | 0.0001 | 0.0085 | 0.0226 | 0.0352 | 0.0302 |
| Pasture*Period | | | 0.1552 | 0.2725 | 0.8448 | 0.0809 | 0.5472 | 0.7643 |

Notes.
[1] Due to differences in the number of sampling time each year, data were analyzed by year.
SEM, standard error mean.
The results in the table are presented as least square means.
a,b,c, the difference between the mean values expressed with different letters in the same treatment group is statistically significant.

Although there was no statistical difference in forage production between them in the second year, oats showed a better response to the high rainfall compared to triticale.

## Nutritive value of cereal forages

The yearly changes in nutritive value of cereal pastures were presented in Table 4. All nutritive value parameters, except for NDF and ADL, varied over the years. In the first year of the study, the highest DMD and ME values were obtained due to the lowest ADF content ($P < 0.05$). There were no significant differences in the nutritive value parameters among the cereal pasture species ($P > 0.05$). The CP contents of oats and triticale were 128.9 g kg$^{-1}$ DM and 123.1 g kg$^{-1}$ DM, respectively, and no significant difference was found between the species in terms of the CP.

The nutritive value of cereal pastures by periods was presented in Tables 5 and 6. The CP contents of cereal pasture species decreased with advancing plant growth stages (Table 5). During the grazing period, the CP contents of oats and triticale showed a similar pattern

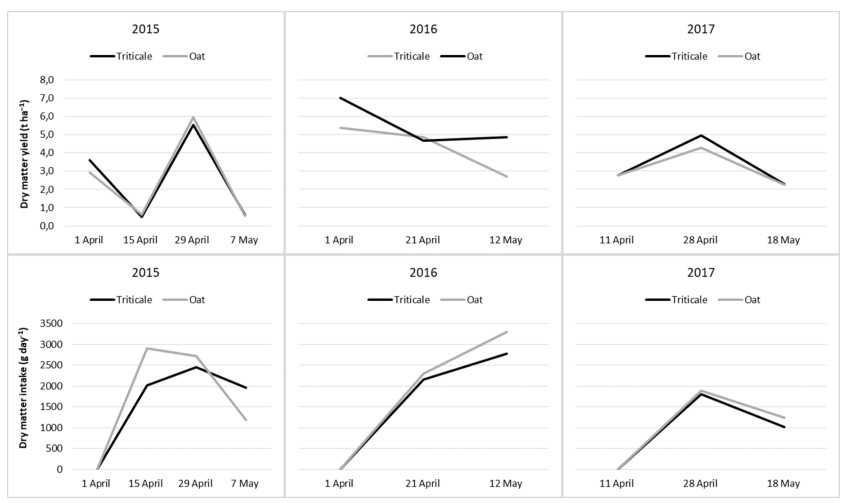

**Figure 1** Changes in dry matter yield and dry matter intake from cereal pastures during the grazing period.

**Table 4 Nutritive value of cereal pastures.[1]**

| Variable | Year | | | Pasture | | P Values | | |
|---|---|---|---|---|---|---|---|---|
| | 2015 | 2016 | 2017 | Oat | Triticale | Year | Pasture | Year*Pasture |
| CP | 101.7[c] ± 4.85 | 127.9[b] ± 5.60 | 148.7[a] ± 5.60 | 128.9 ± 4.38 | 123.1 ± 4.38 | 0.0001 | 0.3528 | 0.4936 |
| Ash | 121.4[b] ± 5.67 | 121.9[b] ± 6.55 | 146.7[a] ± 6.55 | 131.7 ± 5.12 | 128.3 ± 5.12 | 0.0092 | 0.6411 | 0.4689 |
| NDF | 517.6 ± 15.24 | 531.5 ± 17.60 | 521.0 ± 17.60 | 511.9 ± 13.76 | 534.9 ± 13.76 | 0.8307 | 0.2436 | 0.8964 |
| ADF | 185.7[c] ± 11.28 | 274.0[b] ± 13.02 | 337.7[a] ± 13.02 | 261.3 ± 10.18 | 270.2 ± 10.18 | 0.0001 | 0.5399 | 0.9771 |
| ADL | 51.8 ± 3.15 | 47.3 ± 3.64 | 40.7 ± 3.64 | 46.3 ± 2.85 | 46.8 ± 2.85 | 0.0787 | 0.9067 | 0.2218 |
| DMD | 770.3[a] ± 7.94 | 708.1[b] ± 9.17 | 663.2[c] ± 9.17 | 717.0 ± 7.17 | 710.8 ± 7.17 | 0.0001 | 0.5399 | 0.9771 |
| ME | 12.1[a] ± 0.15 | 10.9[b] ± 0.17 | 10.1[c] ± 0.17 | 11.1 ± 0.14 | 11.0 ± 0.14 | 0.0001 | 0.5399 | 0.9771 |

**Notes.**
CP, crude protein; NDF, neutral detergent fibre; ADF, acid detergent fibre; ADL, acid detergent lignin; DMD, in vitro true dry matter digestibility; ME, metabolisable energy.
[1]ME, MJ kg$^{-1}$ DM; others, g kg$^{-1}$ DM.
The results in the table are presented as least square means ± standard errors.
a,b,c, the difference between the mean values expressed with different letters in the same treatment group is statistically significant.

of change (Fig. 2) and therefore, the difference was insignificant ($P > 0.05$). At the start of grazing, the CP content of the forage was highest, but it decreased thereafter. Only in 2017, the CP contents slightly increased from April 11 to April 29, though not significantly, and then decreased significantly later on (Fig. 2).

The variation in the ash content during the grazing period differed across the years. The highest ash content was observed at the end of the grazing period in 2015 and 2016, while in 2017, it was observed at the beginning of the grazing period ($P < 0.05$). Although there were no significant differences in the average ash content of cereal pasture species in 2015 and 2017, triticale had a higher ash content than oats in 2016 (Table 5). During the three years of the study, the variation in the ash content in oats and triticale exhibited similar trends throughout the grazing period (Fig. 2).

**Table 5  Nutritive value (crude protein, ash, in vitro true dry matter digestibility, metabolisable energy) of cereal pastures by periods.[1][*]**

| Sampling time | | | CP | | | Ash | | | DMD | | | ME | | |
| --- | --- | --- | --- | --- | --- | --- | --- | --- | --- | --- | --- | --- | --- | --- |
| 2015 | 2016 | 2017 | 2015 | 2016 | 2017 | 2015 | 2016 | 2017 | 2015 | 2016 | 2017 | 2015 | 2016 | 2017 |
| **Period** | | | | | | | | | | | | | | |
| 1 April | 1 April | 11 April | 123.0 a | 147.9 a | 156.5 a | 136.6 a | 110.0 b | 190.2 a | 775.4 a | 744.3 a | 653.4 b | 12.2 a | 11.6 a | 9.9 b |
| 15 April | 21 April | 28 April | 105.6 b | 116.4 b | 174.2 a | 91.3 c | 117.5 b | 138.2 b | 780.8 a | 704.6 b | 738.3 a | 12.3 a | 10.8 b | 11.5 a |
| 29 April | 12 May | 18 May | 84.8 c | 119.3 b | 115.3 b | 114.5 b | 138.3 a | 111.7 c | 761.1 b | 675.5 c | 598.0 c | 12.0 b | 10.3 c | 8.8 c |
| 7 May | – | – | 92.9 c | – | – | 143.1 a | – | – | 763.9 b | – | – | 11.9 b | – | – |
| SEM | | | 3.684 | 5.069 | 7.399 | 4.391 | 3.401 | 7.139 | 3.077 | 2.312 | 3.675 | 0.058 | 0.044 | 0.069 |
| **Pasture** | | | | | | | | | | | | | | |
| Oat | | | 101.1 | 128.9 | 156.8 | 125.1 | 117.2 b | 152.8 | 772.6 | 712.8 a | 665.6 | 12.1 | 11.0 a | 10.1 |
| Triticale | | | 102.0 | 126.8 | 140.6 | 117.6 | 126.7 a | 140.6 | 768.0 | 703.4 b | 660.9 | 12.0 | 10.8 b | 10.0 |
| SEM | | | 2.605 | 4.139 | 6.041 | 3.105 | 2.783 | 5.829 | 2.176 | 1.888 | 3.001 | 0.041 | 0.036 | 0.057 |
| **P-value** | | | | | | | | | | | | | | |
| Pasture | | | 0.8140 | 0.7258 | 0.0874 | 0.1137 | 0.0374 | 0.1690 | 0.1568 | 0.0056 | 0.2871 | 0.1568 | 0.0056 | 0.2871 |
| Period | | | 0.0001 | 0.0024 | 0.0007 | 0.0001 | 0.0004 | 0.0001 | 0.0013 | 0.0001 | 0.0001 | 0.0013 | 0.0001 | 0.0001 |
| Past.*Per. | | | 0.0967 | 0.1885 | 0.1944 | 0.7479 | 0.0949 | 0.3325 | 0.0340 | 0.3321 | 0.7678 | 0.0340 | 0.3321 | 0.7678 |

Notes.

CP, crude protein; DMD, in vitro true dry matter digestibility; ME, metabolisable energy; SEM, standard error mean.

[1] ME, MJ kg$^{-1}$ DM; others, g kg$^{-1}$ DM.

[*] Due to differences in the number of sampling time each year, data were analyzed by year.

The results in the table are presented as least square means.

a,b,c, the difference between the mean values expressed with different letters in the same. Column is statistically significant.

In all years, the DMD and ME values decreased towards the end of the grazing period ($P < 0.05$). In 2015 and 2017, the difference in DMD and ME values between oats and triticale was insignificant ($P > 0.05$), whereas in 2016, the DMD and ME value of oats was higher than those of triticale ($P < 0.05$). The changes in the DMD and ME of oats and triticale during the grazing period followed similar trends in 2016 and 2017, but not in 2015 (Fig. 2).

The NDF, ADF and ADL values increased as the grazing period progressed over the years ($P < 0.05$). In 2015 and 2017, the NDF content of triticale was higher than that of oats, while in 2016, the ADF and ADL contents of triticale were higher than those of oats (Table 6). Over the entire duration of this study, the changes in cell wall content of oats and triticale during the grazing period exhibited comparable trends, except for ADF changes in 2015 and ADL changes in 2016 (Fig. 2).

## Mineral content of cereal forages

The yearly changes in the mineral content of cereal pastures were presented in Table 7. The effect of the year on the mineral content of the cereal pasture species was insignificant ($P > 0.05$), except for P and Fe. While there was no significant difference in the content of other minerals ($P > 0.05$), the P content of triticale was higher than that of oats ($P < 0.05$).

The macromineral and micromineral content of cereal pastures by periods were presented in Tables 8 and 9, respectively. During the grazing period, the P content of cereal pasture species decreased, with significant reductions noticeable from the end of April onwards. In 2016, the average P content of triticale was higher than that of oats
**Table 6  Nutritive value (neutral detergent fibre, acid detergent fibre, acid detergent lignin) of cereal pastures by periods.[1*]**

| Sampling time | | | NDF | | | ADF | | | ADL | | |
|---|---|---|---|---|---|---|---|---|---|---|---|
| 2015 | 2016 | 2017 | 2015 | 2016 | 2017 | 2015 | 2016 | 2017 | 2015 | 2016 | 2017 |
| | | | | | **Period** | | | | | | |
| 1 April | 1 April | 11 April | 445.3 c | 458.1 c | 455.1 b | 178.4 b | 222.6 c | 351.7 b | 56.1 ab | 40.1 c | 42.5 b |
| 15 April | 21 April | 28 April | 480.9 b | 549.3 b | 462.7 b | 170.8 b | 278.9 b | 231.1 c | 42.0 c | 47.0 b | 15.0 c |
| 29 April | 12 May | 18 May | 578.0 a | 587.2 a | 645.2 a | 198.7 a | 320.3 a | 430.4 a | 46.3 bc | 54.8 a | 64.5 a |
| 7 May | – | – | 566.3 a | – | – | 194.7 a | – | – | 62.9 a | – | – |
| SEM | | | 9.530 | 4.318 | 9.561 | 0.437 | 3.284 | 5.220 | 4.515 | 0.653 | 2.014 |
| | | | | | **Pasture** | | | | | | |
| Oat | | | 503.4 b | 526.6 | 505.7 b | 182.4 | 267.3 b | 334.3 | 56.3 | 44.3 b | 38.4 |
| Triticale | | | 531.8 a | 536.4 | 536.3 a | 188.9 | 280.6 a | 341.1 | 47.3 | 50.3 a | 42.9 |
| SEM | | | 6.739 | 3.526 | 7.807 | 3.090 | 2.682 | 4.262 | 3.192 | 0.534 | 1.644 |
| | | | | | **P-value** | | | | | | |
| Pasture | | | 0.0100 | 0.0783 | 0.0195 | 0.1568 | 0.0056 | 0.2871 | 0.0646 | 0.0001 | 0.0837 |
| Period | | | 0.0001 | 0.0001 | 0.0001 | 0.0013 | 0.0001 | 0.0001 | 0.0228 | 0.0001 | 0.0001 |
| Past.*Per. | | | 0.1598 | 0.1609 | 0.1253 | 0.0340 | 0.3321 | 0.7678 | 0.3114 | 0.0353 | 0.1612 |

Notes.
NDF, neutral detergent fibre; ADF, acid detergent fibre; ADL, acid detergent lignin; SEM, standard error mean.
[1] g kg$^{-1}$ DM.
*Due to differences in the number of sampling time each year, data were analyzed by year.
The results in the table are presented as least square means.
a,b,c, the difference between the mean values expressed with different letters in the same column is statistically significant.

**Table 7  Mineral content of cereal pastures.**

| Variable | Year | | | Pasture | | P values | | |
|---|---|---|---|---|---|---|---|---|
| | 2015 | 2016 | 2017 | Oat | Triticale | Year | Pasture | Year*Pasture |
| P | 2.32$^b$ ± 0.14 | 2.88$^a$ ± 0.17 | 1.93$^b$ ± 0.17 | 2.19$^b$ ± 0.13 | 2.57$^a$ ± 0.13 | 0.0007 | 0.0430 | 0.1834 |
| K | 16.20 ± 1.17 | 16.67 ± 1.35 | 13.50 ± 1.35 | 15.69 ± 1.06 | 15.23 ± 1.06 | 0.2035 | 0.7589 | 0.4525 |
| Ca | 5.03 ± 0.39 | 5.18 ± 0.45 | 4.70 ± 0.45 | 5.09 ± 0.35 | 4.85 ± 0.35 | 0.7435 | 0.6366 | 0.4099 |
| Mg | 2.28 ± 0.46 | 2.38 ± 0.53 | 1.47 ± 0.53 | 1.88 ± 0.41 | 2.20 ± 0.41 | 0.4001 | 0.5920 | 0.9478 |
| Fe | 0.54$^b$ ± 0.18 | 0.30$^b$ ± 0.21 | 1.75$^a$ ± 0.21 | 0.91 ± 0.16 | 0.81 ± 0.16 | 0.0001 | 0.6580 | 0.9302 |
| Cu | 0.04 ± 0.003 | 0.03 ± 0.004 | 0.03 ± 0.004 | 0.03 ± 0.003 | 0.03 ± 0.003 | 0.3851 | 0.8881 | 0.1901 |
| Mn | 0.07 ± 0.011 | 0.05 ± 0.013 | 0.09 ± 0.013 | 0.07 ± 0.01 | 0.06 ± 0.01 | 0.0872 | 0.4615 | 0.0989 |

Notes.
P, Phosphorus (g kg$^{-1}$ DM); K, Potassium (g kg$^{-1}$ DM); Ca, Calcium (g kg$^{-1}$ DM); Mg, Magnesium (g kg$^{-1}$ DM); Fe, Iron (mg kg$^{-1}$ DM); Cu, Zinc (mg kg$^{-1}$ DM); Mn, Manganese (mg kg$^{-1}$ DM).
The results in the table are presented as least square means ± standard errors.
a,b, the difference between the mean values expressed with different letters in the same treatment group is statistically significant.

($P < 0.05$). During the grazing period, the P content of oats and triticale exhibited similar trends, except for changes in 2016 (Fig. 3). At the end of April in each of the three years, the lowest K content was observed. However, in May 2015, the K content increased. There

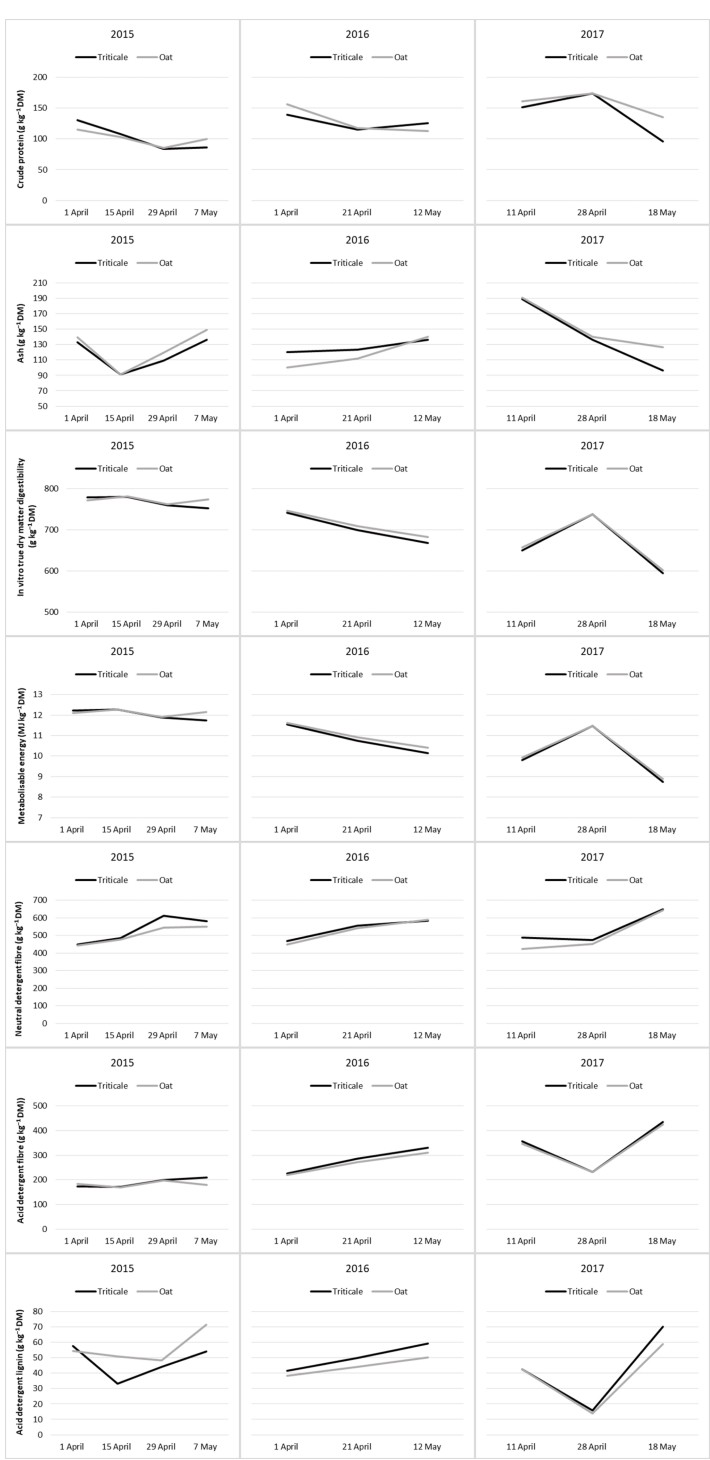

**Figure 2  Changes in nutritive values of cereal pastures during the grazing period.**

**Table 8 Macromineral content of cereal pastures by periods.[1][*]**

| Sampling time | | | Phosphorus | | | Potassium | | | Calcium | | | Magnesium | | |
|---|---|---|---|---|---|---|---|---|---|---|---|---|---|---|
| 2015 | 2016 | 2017 | 2015 | 2016 | 2017 | 2015 | 2016 | 2017 | 2015 | 2016 | 2017 | 2015 | 2016 | 2017 |
| **Period** | | | | | | | | | | | | | | |
| 1 April | 1 April | 11 April | 3.25 a | 3.09 a | 2.20 a | 19.49 a | 18.34 ab | 17.25 a | 6.38 | 5.59 | 6.40 a | 1.96 b | 1.22 | 2.18 a |
| 15 April | 21 April | 28 April | 2.35 b | 3.32 a | 2.00 a | 18.96 a | 19.70 a | 15.06 a | 4.44 | 5.79 | 4.09 b | 0.94 b | 1.33 | 1.07 b |
| 29 April | 12 May | 18 May | 1.29 c | 2.23 b | 1.59 b | 10.01 b | 11.98 b | 8.19 b | 3.92 | 4.15 | 3.62 b | 5.02 a | 4.60 | 1.15 b |
| 7 May | – | – | 2.39 b | – | – | 16.34 ab | – | – | 5.36 | – | – | 1.18 b | – | – |
| SEM | | | 0.193 | 0.164 | 0.111 | 2.184 | 2.439 | 0.979 | 0.810 | 0.524 | 0.679 | 0.795 | 1.118 | 0.137 |
| **Pasture** | | | | | | | | | | | | | | |
| Oat | | | 2.20 | 2.44 b | 1.91 | 17.21 | 15.55 | 14.31 | 5.42 | 4.82 | 5.02 | 1.99 | 2.27 | 1.39 |
| Triticale | | | 2.44 | 3.32 a | 1.94 | 15.19 | 17.80 | 12.69 | 4.64 | 5.53 | 4.39 | 2.56 | 2.49 | 1.54 |
| SEM | | | 0.136 | 0.134 | 0.090 | 1.544 | 1.991 | 0.799 | 0.573 | 0.428 | 0.555 | 0.562 | 0.913 | 0.112 |
| **P-value** | | | | | | | | | | | | | | |
| Pasture | | | 0.2350 | 0.0010 | 0.7970 | 0.3717 | 0.4435 | 0.1830 | 0.3503 | 0.2667 | 0.4389 | 0.4852 | 0.8682 | 0.3524 |
| Period | | | 0.0001 | 0.0019 | 0.0091 | 0.0307 | 0.0046 | 0.0002 | 0.1983 | 0.1015 | 0.0348 | 0.0097 | 0.0977 | 0.0003 |
| Past.*Per. | | | 0.1000 | 0.0433 | 0.3342 | 0.9974 | 0.9485 | 0.2425 | 0.3655 | 0.7178 | 0.9271 | 0.5869 | 0.9405 | 0.0636 |

**Notes.**
[1] g kg$^{-1}$ DM.
*Due to differences in the number of sampling time each year, data were analyzed by year.
SEM, standard error mean.
The results in the table are presented as least square means.
a,b,c, the difference between the mean values expressed with different letters in the same column is statistically significant.

**Table 9 Micromineral contents of cereal pastures by periods.[1][*]**

| Sampling time | | | Iron | | | Zinc | | | Manganese | | |
|---|---|---|---|---|---|---|---|---|---|---|---|
| 2015 | 2016 | 2017 | 2015 | 2016 | 2017 | 2015 | 2016 | 2017 | 2015 | 2016 | 2017 |
| **Period** | | | | | | | | | | | |
| 1 April | 1 April | 11 April | 0.72 | 0.23 | 3.03 a | 0.04 | 0.03 | 0.04 a | 0.13 a | 0.06 | 0.18 a |
| 15 April | 21 April | 28 April | 0.39 | 0.27 | 1.00 b | 0.04 | 0.04 | 0.02 b | 0.04 bc | 0.05 | 0.04 b |
| 29 April | 12 May | 18 May | 0.48 | 0.40 | 1.21 b | 0.04 | 0.03 | 0.04 a | 0.03 c | 0.05 | 0.05 b |
| 7 May | – | – | 0.59 | – | – | 0.03 | – | – | 0.06 b | – | |
| SEM | | | 0.126 | 0.082 | 0.531 | 0.007 | 0.004 | 0.003 | 0.007 | 0.011 | 0.012 |
| **Pasture** | | | | | | | | | | | |
| Oat | | | 0.59 | 0.30 | 1.86 | 0.04 | 0.03 | 0.03 b | 0.07 | 0.08 a | 0.08 |
| Triticale | | | 0.50 | 0.30 | 1.64 | 0.03 | 0.03 | 0.04 a | 0.06 | 003 b | 0.10 |
| SEM | | | 0.089 | 0.067 | 0.434 | 0.005 | 0.003 | 0.003 | 0.005 | 0.009 | 0.010 |
| **P-value** | | | | | | | | | | | |
| Pasture | | | 0.4783 | 0.9619 | 0.7332 | 0.2287 | 0.8501 | 0.0457 | 0.2673 | 0.0023 | 0.0852 |
| Period | | | 0.3069 | 0.3300 | 0.0418 | 0.4222 | 0.5332 | 0.0010 | 0.0001 | 0.7788 | 0.0001 |
| Past.*Per. | | | 0.7209 | 0.5621 | 0.9138 | 0.2242 | 0.2084 | 0.0281 | 0.7410 | 0.6043 | 0.0041 |

**Notes.**
[1] g kg$^{-1}$ DM.
*Due to differences in the number of sampling time each year, data were analyzed by year.
SEM, standard error mean.
The results in the table are presented as least square means.
a,b,c, the difference between the mean values expressed with different letters in the same column is statistically significant.

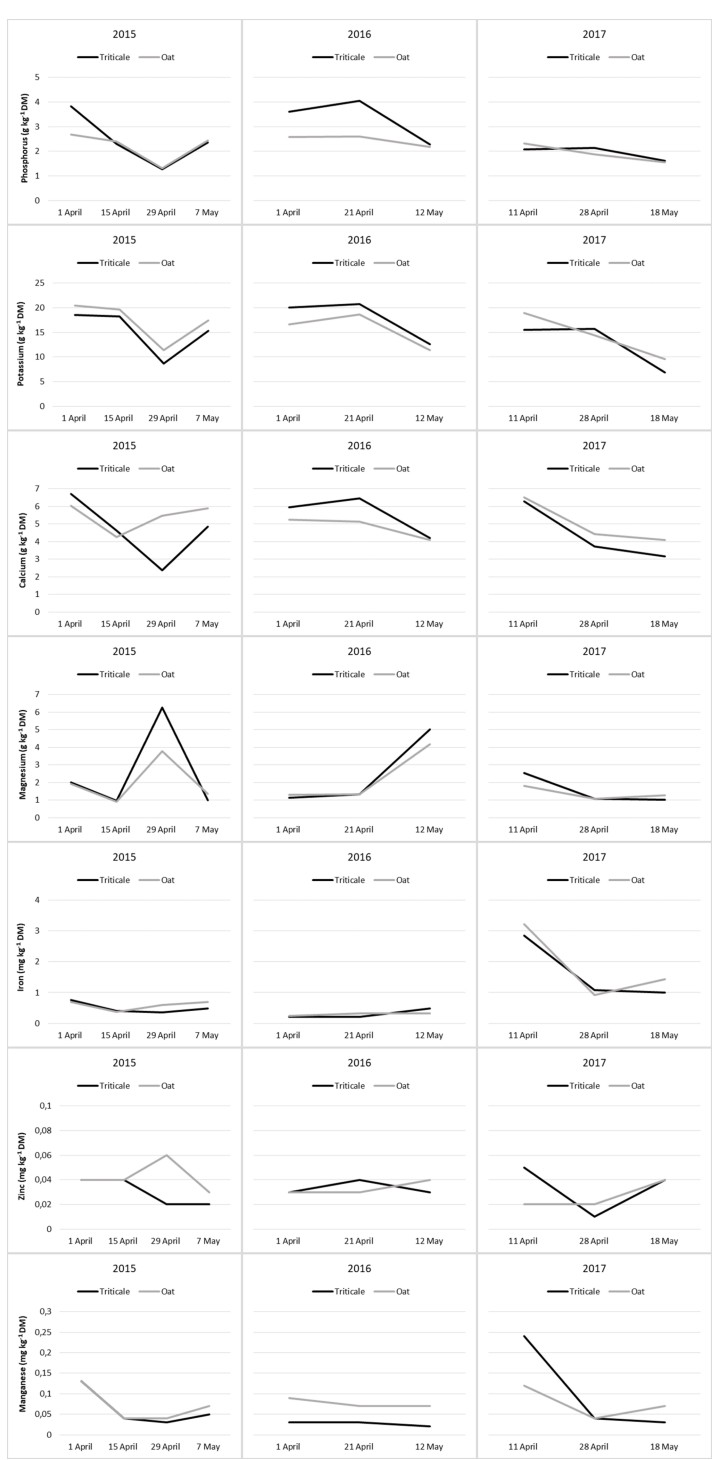

**Figure 3  Changes in mineral content of cereal pastures during the grazing period.**

was no significant difference between oats and triticale for K contents ($P > 0.05$). During the grazing period, the K content of oats and triticale exhibited similar trends (Fig. 3).

Ca contents of cereal pasture species showed a decrease from the beginning to the end of the grazing period, and the change in Ca content during the grazing process was significant in the last year (Table 8). Over the three years, the changes in Ca content of oats and triticale during the grazing period exhibited comparable trends (Fig. 3). During the grazing period, the change in Mg content of cereal pasture species was significant in 2015 and 2017 ($P < 0.05$), but insignificant in 2016 ($P > 0.05$). Over the years, there was no difference in Mg content between oats and triticale ($P > 0.05$), and they showed similar changes during the grazing period (Table 8 and Fig. 3).

In the first two years, the Fe content of the cereal pasture species did not show significant changes during the grazing season ($P > 0.05$). However, in the last year, a significant decrease was recorded from the beginning to the end of the grazing period ($P < 0.05$). For all three years, the Fe content of oats and triticale was similar ($P > 0.05$), and they showed similar changes during the grazing period (Table 8 and Fig. 3). During the grazing period, the change in Zn content of cereal pasture species was insignificant in 2015 and 2016 ($P > 0.05$), but significant in 2017 ($P < 0.05$). In 2017, the average Zn content of triticale was higher than that of oats ($P < 0.05$). During the grazing period, the Zn content of oats and triticale exhibited similar trends, except for changes in 2017 (Fig. 3). In 2015, the Mn content was high at the beginning and end of the grazing period, and lower in the middle. In 2016, the Mn content did not show significant changes during the grazing period, whereas in 2017, it was high at the beginning and then decreased. In 2016, the Mn content of oats was higher than that of triticale ($P < 0.05$). During the grazing period, the Mn content of oats and triticale exhibited similar trends, except for changes in 2017 (Fig. 3).

## DISCUSSION

### Dry matter yield and dry matter intake

The growth of plants and consequently their yield are associated with genetic potential and environmental factors (*Mathan, Bhattacharya & Ranjan, 2016*), and this relationship varies over the years (*Conaghan et al., 2008*). Genotype and environment interactions affect not only forage yield but also its quality (*Vogel, Reece & Nichols, 1993*). There is a close relationship between annual rainfall and temperature (*Pinchak et al., 1996*; *Izaurralde et al., 2011*; *Moore et al., 2021*) and the produced biomass. The rainfall between January and May of 2016 (394.3 mm) being 72.7 mm more than in 2015 and 92.1 mm more than in 2017 significantly increased the forage yield of that year. Additionally, the average temperatures during March to May (13.0 °C in 2015, 14.8 °C in 2016, and 12.8 °C in 2017) were much more suitable for the growth of oats and triticale in 2016, supporting this yield increase (Tables 1 and 2). This is because the growth rates of these plants also increase with temperatures up to 15 °C (*Stichler & Livingston, 1998*). In this study, there was no significant difference in forage yields between triticale and oats (Table 2). The lack of difference between these genotypes grown under the same conditions can be attributed to their similar ecological requirements. The average DMY determined for oats in this study

was consistent with the value reported by *Keles et al. (2016)*, but higher than those reported by *Contreras-Govea & Albrecht (2006)* and *Zhang et al. (2023)*. Similarly, the average DMY for triticale was in line with the values reported by *Keles et al. (2016)* and *Coblentz, Ottman & Kieke (2022)*, yet exceeded those reported by *Coblentz et al. (2018)* and *Akbağ (2022)*. The forage yield of cereal pastures can vary depending on the regions where they are grown, influenced by factors such as climate, grazing management, irrigation, and soil fertility, as well as characteristics like plant density and growth rate (*Phillips et al., 2021*).

The amount of forage consumed by grazing animals in pastures typically varies depending on biomass, the nutritional value of the forage, the physiology and weight of the animal, grazing capacity and grazing system, as well as climate and soil factors (*Papadopoulos, Kunelius & Fredeen, 1993*; *Gordon, 2000*; *Vazquez & Smith, 2000*; *Decruyenaere, Buldgen & Stilmant, 2009*; *Aubé, 2020*). As forage production has increased, forage consumption has also risen (*Assefa & Ledin, 2001*). Hence, in 2016, when forage yield peaked, sheep had the highest DMI. Conversely, in 2017, when forage yield was at its lowest, sheep consumption of forage also dropped to its minimum. The average DMI of oats and triticale in the study was similar (Table 2).

Since climate is one of the most important factors in the growth and the production of plants (*Hatfield & Prueger, 2015*; *Akmarov, Rysin & Knyazeva, 2022*), rainfall and temperature have been decisive in the change in forage yield during the grazing season. In 2015, DMY showed a different variation compared to other years (Table 3). In this year, DMY decreased between April 1st and April 15th and between April 29th and May 7th. Cool-season cereals, such as oats and triticale, generally start growing above 4 °C (*Lemus, 2017*). Therefore, despite low temperatures, the long vegetation period (approximately 5 months) was sufficient for the development of crops sown in autumn and ensured high forage yield at the beginning of grazing. This production level was also expected to be reached before starting grazing. During the two weeks following April 1st, temperatures remained below average, reducing production. In contrast, the rise in temperatures in the second half of April, along with approximately 80 mm of rainfall prior, triggered rapid plant growth. In May, reduced rainfall and increased temperatures suppressed vegetative growth both in 2015 and other years. The highest yields reached on April 29th and 28th in 2015 and 2017 were related to sufficient moisture and optimal temperatures for oats and triticale before these dates. In terms of animal feed consumption, the quality of the forage is as important as its quantity (*Coleman, 2005*). The decrease in DMI in May of 2015 and 2017 was due to the decline in yield and quality of the forages (Table 3). In spring, plants have more leaf tissue, resulting in higher forage quality (*Ball et al., 2001*). For these reasons, grazing sheep consumed the most forage during this month. In May, however, with the weather warming up and precipitation decreasing, cool-season cereals including triticale and oats transitioning from the vegetative to the generative phase resulted in reduced formation of additional leaves and stems (*Virkajärvi, 2006*), leading to a decrease in forage production. Furthermore, as plants enter generative maturity, the CP content decreases while cell wall components increase, reducing the nutritional value and digestibility of the forage (*Nelson & Moser, 1994*; *Buxton, 1996*). High-quality pasture offers herbage that is low in fiber and rich in protein. A reduction in protein availability leads to a decrease

in sheep's herbage consumption (*Stojanović et al., 2016*). As plants mature and become more fibrous, forage intake drops significantly. The potential for intake decreases while the NDF concentration rises as the plants mature. This is because NDF is more difficult to digest than the non-fibrous components of forage (*Ball et al., 2001*). Due to changes in the quantity and quality of forage, DMI decreased during the grazing period in this study, except 2016 (Table 3 and Fig. 1). The amount of forage consumed was proportional to forage production; therefore, in the second year, the highest DMI was recorded because the DMY was greater than in other years, and the yield of the oat pasture was higher than that of the triticale pasture (Fig. 1).

## Nutritive value of cereal forages

Although the nutritive values of cereal pasture species showed significant annual variations, they did not change between species (Table 4). The average CP value determined for oats in this study was consistent with the values reported by *Keles et al. (2016)* and *Zhang et al. (2023)*, but lower than those reported by *Contreras-Govea & Albrecht (2006)*, and higher than those reported by *Abdelraheem et al. (2023)*. Similarly, the average CP value determined for triticale was consistent with the values reported by *Keles et al. (2016)*, but lower than those reported by *Coblentz et al. (2018)* and *Lauriault et al. (2022)*. A significant portion of the proteins found in plants is located in the cell's protoplasm, with very little protein (extensin) present in the cell wall (*Taiz & Zeiger, 2006*; *Hatfield et al., 2007*). As plants mature, the number of mature cells with thicker walls increases, leading to a decrease in the protoplasm and, consequently, the CP content. Therefore, there is abundant research showing that as plants mature, the ratio of CP decreases (*George & Bell, 2001*; *Akbağ, 2022*; *Zhang et al., 2023*). Additionally, maturation leads to a decrease in the leaf/stem ratio, which also causes a reduction in CP levels. This is because leaves are the organs with the highest protein content in the plant (*Hatfield & Kalscheur, 2020*). Therefore, in this study, a decrease in the CP was observed towards the end of April and in May as stem development increased with rapid plant growth (Table 5). Throughout all the years of this study, the variation in CP content of oats and triticale during the grazing period showed similar patterns (Fig. 2).

Ash, representing the inorganic matter or total mineral content of plants, varies according to the plant's species and variety (*Juknevičius & Sabiene, 2007*), environmental factors, agronomic practices ((*Chen et al., 2009*)) and maturity stage; as the maturity stage progresses, the ash content of plants decreases proportionally ((*Mountousis et al., 2008*; *Imoro, 2020*)). Although ash content varied according to the years, the oats and triticale exhibited similar ash content (Table 4). The average ash content of oats was consistent with the value reported by *Hameed et al. (2020)*, but higher than the values reported by *Keles et al. (2016)* and *Samal et al. (2023)*. In contrast, the ash content of triticale was consistent with the values reported by *Coblentz et al. (2018)*, *Akbağ (2022)*, and *Coblentz, Ottman & Kieke (2022)*, but higher than the value reported by *Keles et al. (2016)*. Minerals regulate biochemical reactions and physiological events in plants and play a critical role in their growth and development. Therefore, minerals are present in higher concentrations during the young stages of plants, where physiological activities are most intense and the demand

is highest. As physiological activity decreases towards maturity, there is a tendency for these levels to decline. In this study, a decrease in ash content was observed as forage matured in 2017, consistent with the findings of *Aydoğan & Demiroğlu Topçu (2022)* and *Akbağ (2022)*. Despite the lack of significant differences in the average ash content of cereal pasture species in 2015 and 2017, triticale had a higher ash content than oats in 2016 (Table 5). During the three years of this study, the ash content variation in oats and triticale showed similar patterns throughout the grazing period (Fig. 2).

The digestibility of forage is influenced by numerous factors, including plant and environmental factors. Although the DMD and ME values of cereal pasture species showed significant annual variations, they did not change between species (Table 4). The average DMD value determined for oats was higher than the values reported by *Baron et al. (2012)*, *Kafilzadeh & Heidary (2013)*, and *Kumar et al. (2015)*. For triticale, the average DMD value was consistent with that reported by *Baron et al. (2012)*, higher than the value reported by *Lyu et al. (2018)*, and lower than the value reported by *Keles (2014)*. The average ME value determined for oats was consistent with the value reported by *Keles et al. (2016)*, but higher than the values reported by *Fulkerson et al. (2008)* and *Abdelraheem et al. (2023)*. In the case of triticale, the ME value was consistent with the values reported by *Keles et al. (2016)* and *Akbağ (2022)*, higher than the value reported by *Fulkerson et al. (2008)*, but lower than the value reported by *Glamočlija et al. (2018)*. The differences in digestibility and ME values among the studies are attributed to variations in cultivars and maturity stages (*Masters & Thompson, 2016*). Digestibility varies with the species of plant and its organs (*Van Soest, 1994*) and is also dependent on the plant's growth stage. Digestibility decreases with maturation (*Moore & Jung, 2001*; *Thorvaldsson, Tremblay & Kunelius, 2007*). Additionally, the rate of digestibility is closely related to temperature, with digestibility decreasing as temperatures rise (*Buxton & Fales, 1994*; *Van Soest, 1994*; *Thorvaldsson, Tremblay & Kunelius, 2007*). Therefore, in this study, both the maturation of plants over time and the increase in air temperatures had an effect on the DMD rates of pastures. The research did not reveal a significant difference between the DMD values of triticale and oat, although changes over time were significant (Table 5). Since there was no significant difference in the cell wall components (NDF, ADF, ADL) and CP levels between triticale and oats, there was also no significant difference in their digestibility. This is because digestibility is dependent on NDF, ADF, and CP levels, having a negative relationship with NDF and ADF, and a positive relationship with CP (*Van Soest, 1994*; *Buxton, 1996*; *Ball et al., 2001*; *Mahyuddin, 2008*; *Zhang et al., 2023*). In all years of this study, ME values decreased towards the end of the grazing period, and in 2016, the ME value for oats was higher than that for triticale (Table 5). Over the years of this study, the changes in DMD and ME of oats and triticale during the grazing period followed similar trends, except for 2015 (Fig. 2).

Among the cell wall components, only ADF varied across the years, while oats and triticale had similar NDF, ADF, and ADL contents (Table 4). In this study, the average NDF, ADF, and ADL values determined for oats were consistent with the results of the study conducted by *Keles et al. (2016)* in a similar environment. The values determined for oats were consistent with the results reported by *Zhang et al. (2023)*, but lower than those reported by *Contreras-Govea & Albrecht (2006)*, and higher than the results of

*Abdelraheem et al. (2023)*. The average NDF, ADF, and ADL values determined for triticale are consistent with the NDF and ADF values reported by *Keles et al. (2016)*, *Coblentz et al. (2018)*, and *Lauriault et al. (2022)*, but lower than those reported by *Contreras-Govea & Albrecht (2006)*, and higher than those reported by *Abdelraheem et al. (2023)*. These differences in the chemical composition of cereal pastures can be explained by variations in variety, growing conditions, soil fertility, fertilization, maturity stages, and environmental factors (*Papachristou & Papanastasis, 1994*; *Paulson et al., 2008*).

In plants, the cell wall thickens with the progression of cell development, increasing its proportion relative to the protoplasm (*Taiz & Zeiger, 2006*; *Hatfield et al., 2007*). Consequently, with the advancement of plant maturity, the amounts of NDF, ADF and ADL increase (*Keles et al., 2016*; *Gökkuş, Birer & Alatürk, 2017*; *Akbağ, 2022*; *Zhang et al., 2023*), leading to a decrease in animals' grazing preferences (*Sharpe, 2019*). Additionally, stems, which have cells with thicker walls and less photosynthetic tissue compared to leaves, contain more cell wall material (*Wilson & Kennedy, 1996*). Therefore, with the increase in the ratio of stems to leaves during maturation, the components of the cell wall such as NDF, ADF and ADL also show an increase (*Villalba, Ates & MacAdam, 2021*). In this study, an increase in the NDF, ADF and ADL was observed towards the end of April and in May as the plants matured. In 2015 and 2017, the NDF content of triticale was higher than that of oats, while in 2016, the ADF and ADL contents of triticale were higher than those of oats (Table 6). Throughout all the years of this study, the variation in cell wall content of oats and triticale during the grazing period exhibited comparable trends, except for ADF changes in 2015 and ADL changes in 2016 (Fig. 2).

## Mineral content of cereal forages

The mineral content of plants varies depending on the plant's family, species, maturity stage, soil type and pH, soil mineral levels, and the availability of minerals to the plant (*Underwood & Suttle, 1999*; *Juknevičius & Sabiene, 2007*; *Khan et al., 2007*; *Marković et al., 2019*). Although the P and Fe content of cereal pasture species showed significant annual variations, the other minerals did not change over the years in this study. The mineral content of oats and triticale were largely similar, with only the P content being higher in triticale (Table 7). The macromineral (K, Ca, P, Mg) values determined for oats in this study were consistent with the values reported by *Mut, Akay & Erbaş (2015)* and *Obour, Holman & Schlegel (2019)*, while the micromineral (Fe, Zn, Mn) values were higher than those reported by *Gill & Omokanye (2016)*. The macromineral values determined for triticale were consistent with the values reported by *Sürmen et al. (2011)* and higher than those reported by *Gill & Omokanye (2016)*; *Gill & Omokanye (2018)*, while the micromineral values were higher than those reported by *Gill & Omokanye (2018)*.

In plants, a large portion of P is located in the protoplasm, where physiological events occur intensively (*Spears, 1994*). Additionally, P is fundamental to the energy systems and nucleic acids in plants (*Taiz & Zeiger, 2006*). These compounds enhance physiological activities. Therefore, the decrease in the P content of forage from the beginning to the end of the grazing period can be attributed to an increase in the P during rapid growth periods, followed by a decrease (Table 8). In 2016, the average P content of triticale was higher

than that of oats. During the grazing period, except for in 2015, the P content of oats and triticale exhibited a decreasing trend, with significant reductions noticeable from the end of April onwards (Fig. 3).

Potassium, like P, plays a significant role in physiological processes and regulates the osmotic potential of cells (Hasanuzzaman et al., 2018). Since K is found in small amounts in the cell wall (Spears, 1994), the increase in the cell wall ratio with the maturation of cells indicates a proportional decrease in this mineral (Kelling et al., 2002). Therefore, the average K contents of triticale and oat have shown a decreasing trend from the beginning to the end of the grazing period (Table 8 and Fig. 3).

Calcium is predominantly located in the hard tissues (cell walls) of plants (Spears, 1994). On the other hand, Ca is essential for the activity of cell membranes (Taiz & Zeiger, 2006) and is one of the most important regulators of plant growth and development (Hepler, 2005). Therefore, a significant amount of Ca is also present in the protoplasm. For this reason, the general trend in the change of Ca content in plants tends to decrease as development progresses (George, Nader & Dunbar, 2001; Del Amor & Marcelis, 2005). Consequently, in this study, the Ca contents of pastures showed a decrease from the beginning to the end of the grazing period, and the change in Ca content during the grazing process was significant in the last year (Table 8 and Fig. 3).

Magnesium plays an active role in fundamental physiological and biochemical processes in plants (Ishfaq et al., 2022), resulting in a majority of it being located in the protoplasm (Spears, 1994). Consequently, a decrease in the amount and activity of protoplasm due to maturation is expected to lead to a decrease in Mg content. Therefore, Lisiewska, Korus & Kmiecik (2003) and Jolaosho et al. (2015) have reported that there is a consistent decrease in the Mg ratio of forage with development. However, while the change in the Mg content of forage in the year 2017 showed a decreasing trend, the changes in other years have been the opposite in this study (Table 8 and Fig. 3).

Iron is associated with enzyme systems in plants and acts as a catalyst in biochemical processes such as photosynthesis, respiration, and nitrogen fixation (Hawkes et al., 1985) and is particularly involved in physiological activities in leaf cells (Fodor, 2024). The amount of Fe in plants is influenced by soil conditions, climate, and changes during the growth period (Macpherson, 2000). A decline in F, which is highly active physiologically, is expected with development. This change has been described by Lisiewska, Korus & Kmiecik (2003) as initially increasing and then decreasing towards maturation. Fardous et al. (2011), however, have determined that the change in the Fe content of forage is insignificant and inconsistent. In this study, the change in Fe content was insignificant in the first two years, but due to the reasons mentioned, it was high at the beginning of grazing and then decreased in the last year (Table 9 and Fig. 3).

Zinc, found in most cell walls (Spears, 1994) and constituting the basic structure of some enzyme systems involved in protein synthesis (McCall, Huang & Fierke, 2000), showed insignificant changes in pastures during the grazing period in 2015 and 2016, but the change was significant in the last year. In the final year, the Zn content was high at the beginning and end of the grazing period but decreased in the middle, although the reason

for this change has not been explained. Additionally, in this year, the Zn content of triticale was higher than that of oats (Table 9 and Fig. 3)

Most of the Mn in plants is found in the cell wall (*Spears, 1994*). It acts as a bridge between adenosine triphosphate (ATP) and enzyme complexes, is involved in the activation of certain enzymes in the tricarboxylic acid (TCA) cycle, and participates in redox processes in photosynthesis (*Mousavi, Shahsavari & Rezaei, 2011*; *Schmidt & Husted, 2019*). The Mn content in plants also depends on the plant species and age, as well as the soil's pH and moisture content (*Rayen et al., 2010*; *Stanković et al., 2015*). Due to a decrease in physiological activity, the general trend of decreasing minerals with plant development also applies to Mn (*Fardous et al., 2011*). This study generally reached the same conclusion. Additionally, in 2016, the Mn content of oats was higher than that of triticale (Table 9 and Fig. 3). Since triticale and oats have similar characteristics, their mineral contents were usually found to be close to each other in this study.

Feeds that are adequate in minerals are efficiently utilized by animals. Mineral imbalance not only affects the utilization of minerals but also impacts the rumen environment, reproduction, and animal health, leading to decreased productivity (*Tripathi & Karim, 2008*). Calcium and P are the most abundant elements in the animal body and play a crucial role in the development and maintenance of the organism (*Albu, Pop & Radu-Rusu, 2012*). A deficiency in Ca during critical growth periods reduces the growth rate of the animal and hinders skeletal development (*Burrow et al., 2020*). The physiological functions of K include maintaining electrolyte balance, enzyme activation, and proper functioning of muscles and nerves (*Kumar et al., 2020*). Magnesium is necessary for energy metabolism, genetic code transmission, and nerve signal transmission (*Kumar & Soni, 2014*). The presence of trace elements is essential for the metabolic processes occurring in the rumen microflora. Iron is required for the synthesis of many enzymes during the proliferation of microorganisms (*Tripathi & Karim, 2008*). Zinc is known to activate various enzymatic reactions (*Kumar & Ram, 2021*). Manganese plays a vital role in antioxidants, immunity, growth, and reproduction of animals (*Bjørklund et al., 2020*). Minerals such as Ca, P, Mg, Zn, and Mn all play a role in the successful management of reproductive processes (*Wilde, 2006*).

In all three years of the study, the sheep consumed more daily dry matter than reported by *NRC (2007)* for ewes with single and twin lambs in late lactation. At this level of consumption, the protein and energy requirements of the sheep were met according to *NRC (2007)*. At the same level of consumption, however, the levels of minerals provided to the sheep exceeded the values defined by *NRC (2007)*. The oat and triticale pastures met all the nutritional needs of the sheep throughout the grazing period.

## CONCLUSIONS

Perennial pastures, which are the most important source of roughage for the nutrition of livestock, are directly or indirectly affected by many factors such as grazing, climate, and soil. Early grazing is one of the main issues related to natural pasture in Türkiye. Farmers who do not have sufficient roughage in early spring start grazing perennial pastures early,

leading to the degradation of vegetation and a decrease in grazing capacity. Cereal pastures are among the most important options for addressing the shortage of quality roughage and resolving grazing pressure in perennial pastures. Cool-season cereals like oats and triticale grow rapidly in early spring and reach grazing maturity much earlier than perennial pastures. In the perennial pastures of the South Marmara Region, where this research was conducted, grazing is recommended to start on April 15–20 based on the growth status of the plants. In this study, oats and triticale pastures reached grazing maturity between April 1–11. This shows that established cereal pastures reach grazing maturity 10–15 days earlier than perennial pastures. Study results indicated that triticale and oat pastures have similar nutritional values, making both suitable for providing adequate feed to fulfill the early spring forage needs of sheep. These pastures can reduce the additional energy and protein needs of sheep, but it is important not to overlook that unbalanced supply of minerals. Imbalances and interactions, like mineral deficiencies, can also lead to mineral nutrition disorders. Therefore, cereal pastures can solve the problem of early grazing and have the advantages of high-quality forage without the need for additional feed for sheep.

## ACKNOWLEDGEMENTS

The authors would like to thank Prof. Ahmet GÖKKUŞ for his contributions throughout the study.

### Funding
This research was funded by the "Scientific and Technological Research Council of Türkiye (TÜBİTAK)", grant number: 214 O 233. The funders had no role in study design, data collection and analysis, decision to publish, or preparation of the manuscript.

### Grant Disclosures
The following grant information was disclosed by the author:
The "Scientific and Technological Research Council of Türkiye (TÜBİTAK)": 214 O 233.

### Competing Interests
The authors declare there are no competing interests.

### Author Contributions
- Hülya Hanoğlu Oral conceived and designed the experiments, performed the experiments, analyzed the data, prepared figures and/or tables, authored or reviewed drafts of the article, and approved the final draft.

### Animal Ethics
The following information was supplied relating to ethical approvals (i.e., approving body and any reference numbers):

The Animal Experiments Ethics Committee of Çanakkale Onsekiz Mart University, protocol number: B.30.2. ÇAÜ.0.05.06-050.04/82, approval date: August 27, 2014

## Data Availability

The raw data is available in the Supplemental File.

## Supplemental Information

Supplemental information for this article can be found online at http://dx.doi.org/10.7717/peerj.17840#supplemental-information.

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
