# Peer review of "Forage yields and nutritive values of oat and triticale pastures for grazing sheep in early spring"

_PeerJ, doi:10.7717/peerj.17840_

## Round 0.1 · original submission · Major Revisions

Reviewers submitted very detailed review reports. You should carefully review your manuscript, especially the experimental design sections, correct any errors, and explain them clearly. In addition to reviewers, I recommend shortening the title of your manuscript. Provide a simple and effective title. A humble suggestion
"Evaluation of Oat and Triticale Artificial Pastures in Terms of Hay Yield, Quality and Reduction of Grazing Pressure"
Provide details in the abstract section, not title.

**Language Note:** The review process has identified that the English language must be improved. PeerJ can provide language editing services - please contact us at [email protected] for pricing (be sure to provide your manuscript number and title). Alternatively, you should make your own arrangements to improve the language quality and provide details in your response letter. – PeerJ Staff

Reviewer 1 ·

Basic reporting

The aim of this research is well-thought-out and points out a common problem for the rangelands in Turkey. However, I did not understand the suggestions related to the research findings. The author should implicate why and how grain pastures are grazed.
The English of the text is quite poor so I suggest the author might use the help of a fluent speaker. Additionally, the author/s should be aware of wordiness.
Generally, proper references were used but some of them are too old and should be changed with the recent studies.
Raw data is clear and the article structure is proper. However, I suggest mineral contents should be excluded from this article. It makes the paper very complicated.
Tables should be edited, especially years should be given above the periods and authors should see the variation among the years, periods, and between the pasture types in terms of forage quality characteristics.
The study is well-organized but the hypothesis should be explained clearly and the results/discussion should be given relevant to the hypothesis.
The title is very important for the readers and the title of this study should be revised. For example "Possibility of Reducing Grazing Pressure in Early Spring via Artificially Established Spring Grain (Oat and Triticale) Pasture"

Experimental design

The experimental design is proper and the research has originality.
However, the statistical procedure for animal measurements should be changed. A repeated measurement design should be used for animal data because sampling was carried out from the same unit at different times. This might change the results.
The statistical procedure is proper for pasture data.

Details were given in additional comments

Validity of the findings

Findings should be re-arranged because they are not written in order, which makes it very complicated to read and understand. I suggest using the order in experimental design for the results belonging to variables.

Details were given in additional comments

Additional comments

Abstract
- Aware of the wordiness in the abstract. The language should be corrected.
- Hay is already a dried material so do not use “dry hay” or “fresh hay” terms (L26, L31, etc.)
- Rangeland, mostly consists of native plant species. So use the term “pasture” in the text (L31, L33, L37, etc.)
- Results and suggestions are not clearly explained in the abstract

Introduction
- In L50, it is better to use the “extensive livestock farming” term instead of “Livestock Farming, in terms of pastures”
- Scientific terms should be corrected. For example in L72-73 “hay production in the pasture stops and the plants turn yellow and dry losing their nutritional value significantly” The terms “stop”, and “turn yellow” are incorrect in terms of science. There are more incorrect terms in the text.
The author used a title indicating the alleviation of the early grazing pressure by creating artificial pasture, but the problem of early grazing is not mentioned in the introduction.
- Language should be corrected in the Introduction

Materials and Methods
- In this section, the detailed characteristics of the experimental area should be given. Where was this experiment established?
- L112, why author use “grass yield” instead of “hay yield”?
- It seems that the author tried to explain the method in “Animal characteristics” section L99-131. The method should be given in the Method section and it should be explained more clearly. I suggest the author use figures instead of wordy and confusing explanations.
- Could the author give a reference to the number of seeds that were sown (20 kg/da oat and 30 kg/da triticale) L134? There are other studies that 11-12 kg/da grain seeds (rye, wheat, triticale) are used for creating artificial pasture (Phillips et al., 2021; Lauriault et al., 2022). Moreover, these amounts are better to be given as kg ha-1.
- Any soil analyses were carried out to determine the fertilizer type and amount? L135-136.
- What is the monthly average temperature? Average of 7 months? L145. I did not understand how climatic data were acquired. Besides Celsius sign (°C) should be corrected L145, 148, 149.
- The author should explain why four animals were used in each plot (1080 m2). In L153-156, the daily requirements of the animals were mentioned but this does not explain the amount of the animals or the width of the plots.
- Method for Metabolic Energy should be given in the “Method” section
The statistical method is proper for pasture data but for animals, a repeated measurement design should be used to examine the differences among animal groups because the same animals were measured over time. This may change the results for the animals.

Results
The author stated sampling was done 4 times in 2015, 3 times in 2016, and 3 times in 2017. It should be explained in the “Method” section why sampling amounts changed in years and when the sampling was done.
- Lines 188-189-190 should be given in the method section, not in the results.
- As I understand from the study results, “fresh hay” should be converted as “fresh forage”, and “dry hay” should be converted as “hay” in the whole text.
- L192-193, I did not understand what is the changing of forage amount according to sampling times.
- L196 the sentence is unfinished “while that of the oat pasture…..” please complete it.
- All yield results should be given as kg ha-1, not kg/da. Please edit the units.
- In result tables, the year should be given above the months.
- Decimals should be given with a point “.” not comma “,”.
- There too much data given in the results, which makes the study results is very complicated. For example, dry matter consumption of the animal should be given only over a “kg sheep period”. Dry matter consumption of the animals within the sampling period or daily consumption averages within the period. Thereby Table 5 should be changed also.
- Results that belong to fresh forage should be excluded. So use only hay yield results (dry hay as it was mentioned in the text) because dry matter is the main characteristic to indicate yield and also to calculate grazing-related information. Results will be more clear by using only one yield characteristic (hay).
- Tables indicating forage quality characteristics should be edited. Readers should see how one character changed among the years, periods, and pasture types in one table. Please edit them (Tables 8, 9, 10, 11, 12, 13). Additionally, there are too many characteristics given in the study results. The author may consider using only yield, digestibility, and energy characteristics by excluding the data of elemental content (Table 11, 12, 13) L345-422 in the results.
- L243 “A significant difference emerged between…” change the word “emerged” to “occurred”. Also in L271, 276,
- In L243, 245, 276, 290…. Why author use “herb” instead of “hay”? Please change them.
- L251-253, I did not understand what are experimental plots. Why variations among the plots were explained? Plots are animals here? This needed to be clarified.
- If the same animals were measured at all samplings, repeated measurement designs should be used for the statistical analysis. Therefore statistical procedure should be edited and this may cause significant change in the results.
- In Table 6, 7 standard error abbreviations should given in English as SE, not SH.
- L 286, “These figures show that….” which figures mentioned?
- For CP results, one decimal is enough for example 9.29 % is 9.3%.
- Give the results in the same order as the ANOVA tables. For example, first, give the results of the years, then pasture types, and then sampling periods. Results are very confusing as it is so please revise.
- L326 “There was non-significant difference…” change to “ The difference between … was not significant…..” There are too many phrases that need to be edited in terms of language.
- Section for Mineral Elements Concentration could be removed L345-422.

Discussion
- L428-429 the given references are too old; more recent studies should be cited.
The first sentences of the discussion should be more impressive. I did not find any explanation for differences in pasture types and hay consumption in the first sentence.
- L433-434 “Additionally, since the plants in this location have more leaf tissues, their hay quality is also high” This sentence does not make sense because a higher leaf ratio means higher forage quality for all locations in the world not for this location only.
- In discussion, results belong to the factors that should be discussed in order. The order in the ANOVA table or statistical design should be followed (i.e. years, pasture types, periods) for each characteristic (hay yield, consumption, quality, etc.)
- L441-442 Poaceae and Fabaceae terms are not suitable here, use terms such as grasses and legumes.
- L441-444 “Furthermore, cool climate poaceae and Fabaceae fodder crops produce small cells having thicker walls when grown under high temperatures and they store less non-structural carbohydrates with producing less digestible hay for animals” Are there any reference to this information?
- L444-445 it is not necessary to discuss the seasonal change in the forage quality of legumes.
- There is not a scientific term for “Poaceae fodder crops”, please revise it in the discussion.
The author gave scientific results in discussion about the variation of hay yield, forage quality, and consumption differences but this information should be discussed with the study results. The discussion here is written much likely a review article. Please revise the discussion section and discuss the results of the study by explaining the reasons for the results.
- Exclude the mineral contents in the discussion too.

Conclusions
- L537 “Yields of hay of grain pastures showed a decrease……” should be changed to “Hay yield of pastures decreased.…..” English must be revised in all text.
- In conclusion, do not give results. A general conclusion and suggestions should be written in this section.
- All conclusion sections should be revised by including suggestions.

* Phillips et al., 2021; H. N., Heins, B. J., Delate, K., & Turnbull, R. (2021). Biomass Yield and Nutritive Value of Rye (Secale cereale L.) and Wheat (Triticum aestivum L.) Forages While Grazed by Cattle. Crops, 1(2), 42-53.
* Lauriault, L. M., Schmitz, L. H., Cox, S. H., Duff, G. C., & Scholljegerdes, E. J. (2022). A comparison of native grass and triticale pastures during late winter for growing cattle in semiarid, subtropical regions. Agronomy, 12(3), 545.

Reviewer 2 ·

Basic reporting

This study employed a randomized complete block design with three replications, grazing 2-3 year old Karacabey Merinos sheep on each experimental plot. It examined the hay yield (both yield and consumed hay amount), hay quality (dry matter ratio, crude protein, NDF, ADF, ADL, digestible organic matter, metabolizable energy, and mineral content), and the body weight and condition scores of the animals grazed on these artificial pastures.
The manuscript clearly reports on the background, objectives, methods, main results, and conclusions of the study. The abstract and introduction provide sufficient background information and significance of the research. The literature review is comprehensive, showcasing the study's novelty and its relationship with existing research.

Experimental design

The experimental design is sound, with the methodology section detailing the trial design, animal management, forage planting, and data collection processes. However, further clarification on the rationale behind the selection of specific forage varieties (such as oats and triticale) and their potential impact on the study results is advised.

Validity of the findings

The results section thoroughly reports on changes in hay yield, quality, and animal weight and condition, with data reasonably interpreted through statistical analysis. Figures and tables clearly present the data, but additional explanations regarding statistical significance and the practical implications of these findings for pasture management are recommended.

Additional comments

1. The overall organization of the manuscript is good, but certain sections (like the discussion) could be further strengthened to more deeply analyze the potential impacts of the study results on mitigating early spring grazing pressure, as well as how grassland management strategies can be adapted to address climate change and variability in forage production.
2. It is suggested that the discussion section explores the adaptability and production efficiency of oat and triticale pastures under different climatic conditions and soil types, as well as their long-term effects on improving animal production performance and welfare.
3. While the manuscript provides a range of useful findings, further depth and breadth could be added to the research by exploring the impact of different grazing densities and management practices (such as rotational grazing) on hay yield and quality.
Overall, this manuscript offers valuable insights into addressing the spring feed gap issue in natural pastures and demonstrates the potential of artificial pastures in mitigating early spring grazing pressure. The authors are advised to consider the above comments to further enhance the manuscript's quality and practicality.

Reviewer 3 ·

Basic reporting

The objective of the research was to cover the feed deficit of natural grasslands during spring by means of cultivated oats and triticale for grazing, determining yield and quality of hay. However, the manuscript is not coherent with the objective nor with the title.
The manuscript has severe deficiencies. The methodology is not correct, as the statistical design does not address the stated objectives; and therefore, the design is not appropriate.
Results are not properly presented.
There are many references over ten years old.
English is deficient and would need extensive editing.

Experimental design

Not adequate.

Validity of the findings

Results are not properly presented.

Additional comments

No additional comments. The manuscript is not acceptable for publication.

Reviewer 4 ·

Basic reporting

The study is very valuable as it is a 3-year field study. The fact that the study has a specific hypothesis also makes it original. However, the fact that what is intended to be explained is not clearly expressed shows that the article needs intensive correction work due to the weakness of the English writing language and the large number of inaccuracies in terminology.

In the Abstract section, the feed expression should be corrected to forage. The background part is also explained very briefly. There are small and capital letter problems in some sentences (28). The expression hay is generally defined for dry forage. The expression green hay needs to be changed.

For Introduction
50-Livestock Farming written in capital letters. This statement should be re-evaluated terminologically. There is no animal production based solely on pasture in the world. Different land types such as rangeland, grassland and meadow are used intensively for grazing or animal production.
52-Sentence should be arranged.
57-Pasture farming terminology?
62- Farm animals->livestock
88-It may not always be right to choose triticale over oats. This statement is stated quite precisely. You need to enrich it with other literature. At the same time, the quality values given are the values of only one study. Apart from these studies, studies conducted in different ecologies may show different results. At the same time, expressions such as discussion should be avoided in the introduction.
104-Statements should be explained in a more academic language.

Experimental design

Focusing on two species in the trial design creates a limitation for the statistical interpretation of the study. An abundance of sources of variation could have enriched the study. Are you sure that the trial design is randomized complete block design?
110-According to what standards were the fence values given for grazing determined? It should be supported by literature.
115- Parcels -> Plots
133- Latin name?
134- Which seeder? Nowadays, it seems that even sowing machines are very important in the work.
140- Soil characteristics without numbers? Which techniques are used for analysis?
172- Forage quality - There is no method in the Ash analysis section. Which machine and which methods were used to analyze the ash? I think that giving only AOAC standards is not appropriate for such journal standards.
The same applies to ADF, NDF and crude protein. It should be explained why mineral substance analyzes are performed. Why mineral matter is so important in animal production and animal trials should be included as a hypothesis.
180- Ankom-> ANKOM
183- Statistic modeling should give wider.
188- Four plant samples? What does it mean?
191- Fresh hay? Correct terminology...

Validity of the findings

208- 262 The trial is a 3-year trial, but significant differences were detected between the years. The reason for this should be explained comprehensively. For example, why did the values ​​decrease in the 2nd year? Why are interactions so important?
266- Didn't ->did not
278- April 1st ... - Although it is considered a high value, it appears to be low. The sentence should be corrected.
280-The sentence does not understand.
291- Which are the others?
Although the values seem to be quite good, they cannot be fully understood due to the weak expression language.
307- It was immediately switched from NDF to ADL. And a single value was explained.
385- Nothing results?
436- Grain expression is not used for forage crops.
441- Latin corrections?
444- What literature supports its accuracy? There are too many decrease and increase expressions.
539- Said?

Additional comments

The study is original and was carried out in the field for 3 years. This type of studies are very valuable in the management of pasture areas. The study can only be published with radical corrections in the writing language and methodology.

---

## Round 0.2 · Minor Revisions

After the remaining minor revisions mentioned by the reviewers, your manuscript should be acceptable.

Reviewer 1 ·

Basic reporting

The aim of this research is well-thought-out and points out a common problem for the rangelands in Turkey and the study is well-designed. The author used proper language and sufficient literature. However, a few corrections are required for the reference list (given in the additional comments). All figures and tables have a professional structure. The given results are relevant to the hypothesis. Raw data is clear and the article structure is proper.
The title may be "Oat and Triticale Pastures for Sheep Grazing in Early Spring" if the authors wish. This is occasional.

Experimental design

The experimental design is proper and the research has originality. The statistical procedure is appropriate. The authors well-defined the research question.

Validity of the findings

All underlying data is provided and the conclusion is well-linked to the research question.

Additional comments

It is understood that the authors successfully revised the manuscript and I think now the paper fits the journal's scope. Only a few minor corrections are required, mostly spelling, and they are given below.

Abstract
L23, 24, 25 "sheep" should be "sheeps"

Reference list
- "Al Jundi A. 2010" is not used in the main text.
- "Ayub, M., .......Sarwar N. 2011" is not used in the text
- "Coblentz WK, ........Cavadini JS. 2018" should be replaced in the list. Put it after "Coblentz WK, Gildersleeve RR. 2014"
- "Taiz and Zeiger" same reference is cited as 2006 and 2008. Please correct.

Reviewer 2 ·

Basic reporting

The research design and experimental methods are clearly described, and the data collection process adheres to ethical standards. The use of triticale and oats as early spring forage crops in the study is significant in alleviating grazing pressure and potentially mitigating feed shortages. However, while the manuscript is clear and professional, certain expressions are imprecise (e.g., lines 23, 77, 121, 128), which may hinder comprehension for international readers. It is recommended to seek assistance from a colleague with strong English skills and familiarity with the field or to use professional editing services to enhance language quality. Additionally, the provision of raw data is commendable, but the supplementary file lacks adequate metadata, reducing the usability of these data for future readers.

Experimental design

The study is an original and high-quality research contribution with a clear and practically significant research question, addressing a knowledge gap. However, the methods section requires more detailed descriptions to enable replication of the experiments. Although the experimental design complies with the journal's scope and requirements, the data analysis could be improved.

Validity of the findings

All primary data have been provided, with high data quality, robust statistics, and appropriate controls. Nevertheless, the study's impact and novelty have not been sufficiently evaluated. It is recommended to emphasize this aspect in the revised manuscript. Additionally, if the data re-analysis can significantly contribute to the literature, meaningful replication should be encouraged.

Additional comments

In summary, while there are issues with language expression and data description, the overall structure and research content of the manuscript are commendable.

Reviewer 4 ·

Basic reporting

The author has taken significant account of the previously mentioned corrections, making the article publishable.

Experimental design

The author has taken significant account of the previously mentioned corrections, making the article publishable.

Validity of the findings

The author has taken significant account of the previously mentioned corrections, making the article publishable.

Additional comments

The author has taken significant account of the previously mentioned corrections, making the article publishable.

---

## Round 0.3 · accepted · Accept

The corrections made are suitable for acceptance of the manuscript. Congratulations